# Greenhouse Gas Emissions and Yield Production from an Organic and Conventional Fertilization on Quinoa

Jorge Alvar-Beltrán [ID], Anna Dalla Marta [ID], Roberto Vivoli, Leonardo Verdi *[ID] and Simone Orlandini [ID]

Department of Agriculture, Food, Environment and Forestry (DAGRI), University of Florence, 50144 Florence, Italy; jorge.alvar@unifi.it (J.A.-B.); anna.dallamarta@unifi.it (A.D.M.); roberto.vivoli@unifi.it (R.V.); simone.orlandini@unifi.it (S.O.)
* Correspondence: leonardo.verdi@unifi.it; Tel.: +39-0552755741

**Abstract:** The high nutritional properties of quinoa have resulted in a production increase worldwide. The resistance to environmental stresses renders this crop suitable for sustainable farming systems. Few studies have examined the impact of different agricultural management strategies and its contribution to climate change. In this work, we quantify soil greenhouse gas (GHG) emissions, in terms of carbon dioxide ($CO_2$), methane ($CH_4$) and nitrous oxide ($N_2O$), and crop productivity (yields and biomass) under conventional (urea) and organic (digestate) fertilization. Significant differences ($p < 0.05$) in $N_2O$ cumulative emissions are reported between digestate (50–100 kg N ha$^{-1}$), urea (50–100 kg N ha$^{-1}$) and the control (0 kg N ha$^{-1}$). Higher cumulative GHG emissions are observed under 100 kg N ha$^{-1}$ of digestate (337.8 kg C ha$^{-1}$ $CO_2$ and 0.23 kg N ha$^{-1}$ for $N_2O$) compared to treatments with lower nitrogen (N) inputs. However, yield and biomass production do not show significant differences ($p > 0.05$) with increasing nutrient application. Hence, this study opens the discussion about the pros and cons of increasing fertilization to improve yields besides providing agricultural extension workers with additional information to promote sustainable quinoa production worldwide.

**Keywords:** digestate; urea; nitrogen; carbon dioxide; methane; nitrous oxide; global warming potential

## 1. Introduction

The use of fertilizers in agriculture are responsible for a large share of anthropogenic emissions [1]. According to the International Panel on Climate Change (IPCC), at global level, 1% of the applied nitrogen (N) in agriculture is lost as nitrous oxide ($N_2O$) emissions [2]. Other studies suggest that $N_2O$ losses can range between 0.03 and 14% of the applied N depending on the regions, soil texture, and crops [1]. It is also estimated that $N_2O$ emissions from agriculture represent approximately 80% of the global anthropogenic emissions. The magnitude of the environmental issue increases when considering organic fertilizers, which are important sources of nutrients sometimes missing in synthetic fertilizers, and when misused, can become large sources of greenhouse gases (GHG) [3,4]. Generally, due to their own composition, liquid-organic fertilizers provide higher $N_2O$ emissions compared to solid-organic fertilizers [1,5]. Organic fertilizers can improve nutrient availability, biodiversity, and microbial activity in the soil. Conversely, as soil organic carbon (C) increases, so does the microbial activity, which consequently accelerates carbon dioxide ($CO_2$) emissions through soil respiration [6]. Recent literature reports a direct effect of organic fertilization, with compost and digestate, on methane ($CH_4$) emission increase [7,8].

Due to its high tolerance to biotic and abiotic stresses, and nutritional properties, the cultivation of quinoa is gaining scientific attention [9]. Much of the scientific efforts have been devoted to the study of quinoa's resilience to water, salinity, temperature, and nutrient stress, but little is known about the linkages between GHG emissions and increasing

fertilization. Hence, the compounded effects of agricultural strategies on improving crop yields represents an area of major interest in agriculture [10]. For quinoa, discrepancies emerge between the actual effect of fertilization on biomass production and seed yields. For example, study [11] observes a positive relationship between yields and N fertilization rates. Similarly, study [12] shows a yield increase of up to 75–100 kg N ha$^{-1}$. With regards to the type of fertilizer, most of the research on quinoa has assessed the effect of synthetic fertilizers, particularly urea ($CO(NH_2)_2$) and ammonium nitrate ($NH_4NO_3$), on quinoa yields. Ref. [11] describes an increase in plant height and seed yield with rising $NH_4NO_3$ rates. The same study reports an increase in yield from 0.9 to 9.2 g plant$^{-1}$ under 0 and 122 kg N ha$^{-1}$, respectively. Similar substantial yield enhancements are displayed when N-fertilization rates are increased from 40 to 120 kg N ha$^{-1}$ [13], as well as with calcium ammonium nitrate ($NH_4NO_3$), with yields increasing from 1.8 to 3.5 Mg ha$^{-1}$ under 0 and 120 kg N ha$^{-1}$, respectively [14].

On the contrary, the authors of [15,16] report little to no differences in terms of yield and biomass production up to 25 kg N ha$^{-1}$. In addition, studies [15–17] suggest that N-fertilization has a minor effect on quinoa performance, requiring approximately 25 kg N ha$^{-1}$ per ton of quinoa seeds produced. Ref. [10] affirm that N splitting is not an advantage for fast-developing species such as quinoa, particularly in terms of seed yield and protein content.

Despite the previous studies, there is still limited information on quinoa's performance under organic crop-systems. Existing studies have assessed different organic fertilizers, such as compost and cow manure, but do not show notable differences in yields [18]. Instead, Ref. [10] report a yield increase with organic N-fertilization (60, 120, and 180 kg N ha$^{-1}$ of slurry), observing a yield of 2.20 Mg ha$^{-1}$ under 180 kg N ha$^{-1}$ of slurry. Similarly, study [19] report 1.20 Mg ha$^{-1}$ (Ayacucho) and 1.70 Mg ha$^{-1}$ (Huancavelica) with 35 Mg ha$^{-1}$ and 54 Mg ha$^{-1}$ of poultry fertilizer (guano), respectively. From the former literature review, we conclude that countless factors are responsible on improving yields and, consequently, require additional scientific attention; in particular, on the linkages between fertilization and yields, and the environmental costs associated with higher agricultural inputs [20]. Lastly, recent studies highlight the differences in GHG emissions because of the type of fertilizer (organic and synthetic), pedoclimatic conditions, and agricultural management strategies [1]. This emphasizes the need to conduct case-specific observations to obtain information about emission dynamics from agricultural systems.

In the present study, digestate and urea are assessed at different rates (0, 50, and 100 kg N ha$^{-1}$) to better understand their effect on quinoa yields, biomass production, and environmental impacts.

## 2. Materials and Methods

### 2.1. Experimental Design and Agricultural Management Strategies

This study was run between May and August 2019 at the experimental field of Istituto Tecnico Agrario Statale (ITAS) (43° 47′ 06″ N and 11° 13′ 06″ E; 40 m.a.s.l.) in Tuscany, Italy. A randomized complete block design (RCBD) was used, divided in two types of fertilizer (urea (U) and digestate (D)) with two different N-levels (50 and 100 kg N ha$^{-1}$) and a control (0 kg N ha$^{-1}$); hereafter referred to as Control, 50D, 100D, 50U, and 100U. Each treatment included three replicates, for a total of 16 plots (Figure 1). Each plot sized $\pm 4$ m$^2$ (2.1 m width and 1.8 m length), with four rows spacing 70 cm and plants distancing 10 cm from each other (15–20 plants m$^{-2}$ and $\pm 75$ plants plot$^{-1}$). Thus, experimental design was: Control (0 kg N ha$^{-1}$); 50 D (50 kg N ha$^{-1}$ as digestate); 100 D (100 kg N ha$^{-1}$ as digestate); 50 U (50 kg N ha$^{-1}$ as urea); 100 U (100 kg N ha$^{-1}$ as urea).

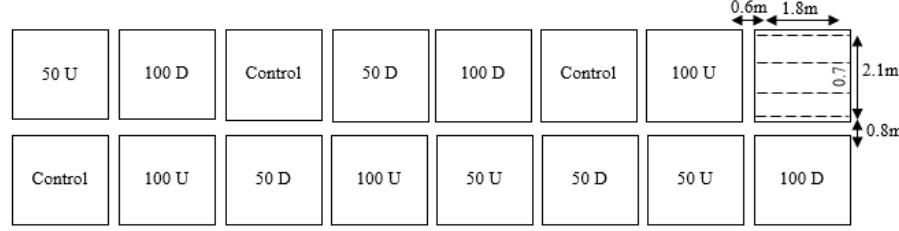

**Figure 1.** Experimental design with different N-fertilization levels: Control (0 kg N ha$^{-1}$); 50 D (50 kg N ha$^{-1}$ as digestate); 100 D (100 kg N ha$^{-1}$ as digestate); 50 U (50 kg N ha$^{-1}$ as urea); 100 U (100 kg N ha$^{-1}$ as urea).

The genotype Titicaca was sown on 8 May and harvested on 26 August, with a total cycle duration of 110 days. Urea (CO(NH$_2$)$_2$) and the liquid fraction of digestate from pig slurries (Table 1) were manually spread in two equal doses, and immediately incorporated into the soil (at 20 cm depth) using a tractor (BCS 710 ACTION). Fertilization was carried out 28 and 49 days after sowing (DAS), coinciding with the stem elongation and flowering phase, respectively. Mechanical weeding was done at the same time as fertilizer application using the same walking tractor. The soil texture in the study field was characterized for being sandy-loam (56% sand, 15% clay, and 29% silt) with a total N content of 0.15 g kg soil$^{-1}$ at 0–20 cm depth.

**Table 1.** Elemental characterization of fertilizers (urea and digestate) used in this experiment.

|  | Units | Urea | Digestate |
|---|---|---|---|
| Organic C | % | - | 3.02 |
| N content total | % | 46 | 0.39 |
| N-NH$_4^+$ | % | - | 0.30 |
| N-NO$_3^-$ | % | - | <0.01 |
| P content total | mL l$^{-1}$ | - | 452 |
| K content total | mL l$^{-1}$ | - | 2457 |
| Dry matter | % | 100 | 1.89 |

*2.2. Crop and Agrometeorological Measurements*

The monitoring of plant's phenology was performed during the entire growing season. Measurements were taken from the two middle rows to avoid any side effects. For the evaluation of crop performance (yield and biomass production), all the plants in the middle rows (38 plants plot$^{-1}$) were sampled at physiological maturity and manually harvested using a sickle. Dry weight of grains and biomass (stems and leaves) were determined by drying samples in an oven at 80 °C for 48 h. To standardize the results, yield (grains) and biomass production (stems and leaves) were converted to kg ha$^{-1}$. Harvest index (%) was calculated as the ratio between yield (grains) and biomass (Table 2). To complete the agrometeorological analysis, weather information (daily average, maximum and minimum temperature, and precipitation) was obtained from both the Regional Hydrological Services of Tuscany (SIR) and the Functional Centre of the Tuscany Region (CFR).

**Table 2.** ANOVA test (±Standard Deviation) for seed yield (kg ha$^{-1}$), aboveground biomass (kg ha$^{-1}$), and harvest index.

| Fertilizer Type | N-Treatment (kg ha$^{-1}$) | Yield (kg ha$^{-1}$) | Biomass (kg ha$^{-1}$) | Harvest Index (%) |
|---|---|---|---|---|
| Control | 0 | 844 ± 126 a | 3188 ± 382 a | 22.4 ± 3.27 a |
| Digestate | 50 | 750 ± 154 a | 3188 ± 382 a | 23.6 ± 3.46 a |
| | 100 | 792 ± 138 a | 3832 ± 856 a | 20.9 ± 2.42 a |
| Urea | 50 | 852 ± 106 a | 3410 ± 338 a | 25.0 ± 3.07 a |
| | 100 | 894 ± 422 a | 4064 ± 2244 a | 22.7 ± 2.22 a |
| Control | | 844 ± 126 a | 3188 ± 318 a | 22.4 ± 3.27 a |
| Digestate | | 772 ± 132 a | 3510 ± 690 a | 22.3 ± 3.40 a |
| Urea | | 874 ± 276 a | 3738 ± 1480 a | 23.9 ± 2.72 a |
| | 0 | 844 ± 126 a | 3188 ± 318 a | 22.4 ± 3.27 a |
| | 50 | 800 ± 130 a | 3298 ± 344 a | 24.3 ± 3.40 a |
| | 100 | 844 ± 286 a | 3948 ± 1524 a | 21.8 ± 2.30 a |

Means that do not share a letter are significantly different from each other ($p < 0.05$).

## 2.3. Greenhouse Gas Measurements

GHG measurements were recorded biweekly from the middle rows during the entire growing season: at 7 DAS (hereafter, M1), 16 DAS (M2), 28 DAS (M3), 33 DAS (M4), 49 DAS (M5), 63 DAS (M6), 77 DAS (M7), 93 DAS (M8), and 110 DAS (M9), respectively. GHG monitoring was performed using 16 static chambers (one per plot) and a portable gas analyzer (XCGM-400, Madur Sensonic). Soil emissions measurements were carried out between the rows avoiding the inclusion of plants inside static chambers. Chambers were assembled following the USDA-ARS GRACEnet Project Protocols [21], as described by [8]. Chambers were made by two parts: chamber lids [8] and chamber collars that were inserted into the soil at 5 cm depth to avoid any mechanical damages to the plant's root system. The portable gas analyzer used non-dispersive infrared (NDIR) technology for the analysis of $CO_2$ (±0 ppm), $CH_4$ (±10 ppm), and $N_2O$ (±1 ppm). Gas concentration inside the chambers (ppm), area (314 cm$^2$), volume (9420 cm$^3$), closing time (one hour), and molecular weight of each gas were considered for computing soil GHG fluxes. Cumulative soil GHG emissions at the end of growing season were expressed as kg C ha$^{-1}$ for $CO_2$ and $CH_4$, and as kg N ha$^{-1}$ for $N_2O$.

For the observations on emissions fluxes and yields analysis, a yield-scaled emission calculation was used for each gas. Yield-scaled emissions were expressed as the net contributes of 1 kg of quinoa grains (kg $CO_2$ eq.) The kg $CO_2$ eq. were obtained from gas-specific Global Warming Potentials (GWP) ($CH_4$ and $N_2O$ with values of 28 and 298, respectively) [22]. The total impact of each treatment was determined as kg $CO_2$ eq. and computed separately for the cumulative sum of each GHG (Table 3).

**Table 3.** Total cumulative GHG emissions during the growing season and daily GHG fluxes.

| Fertilizer Type | N-Treatment (kg ha$^{-1}$) | GHGs during the Growing Season | | | GHGs Average Day$^{-1}$ | | |
|---|---|---|---|---|---|---|---|
| | | CO$_2$ (kg C ha$^{-1}$) | CH$_4$ (kg C ha$^{-1}$) | N$_2$O (kg N ha$^{-1}$) | CO$_2$ (kg C ha$^{-1}$ day$^{-1}$) | CH$_4$ (kg C ha$^{-1}$ day$^{-1}$) | N$_2$O (kg N ha$^{-1}$ day$^{-1}$) |
| Control | 0 | 223.1 ± 16.4 b | 1.73 ± 0.27 | 0.04 ± 0.02 c | 23.3 ± 1.6 b | 0.21 ± 0.03 | 0.003 ± 0.001 c |
| Digestate | 50 | 293.6 ± 20.9 ab | 2.40 ± 0.34 | 0.15 ± 0.01 b | 30.6 ± 2.9 ab | 0.30 ± 0.04 | 0.015 ± 0.001 b |
| | 100 | 337.8 ± 19.7 a | 2.29 ± 0.26 | 0.23 ± 0.01 a | 36.1 ± 2.3 a | 0.28 ± 0.03 | 0.025 ± 0.003 a |
| Urea | 50 | 220.7 ± 32.4 b | 2.23 ± 0.94 | 0.13 ± 0.04 b | 23.4 ± 3.7 b | 0.27 ± 0.10 | 0.012 ± 0.005 b |
| | 100 | 229.1 ± 45.5 b | 2.22 ± 0.66 | 0.11 ± 0.03 b | 23.5 ± 5.1 b | 0.26 ± 0.08 | 0.010 ± 0.003 b |
| Control | | 223.1 ± 16.4 b | 1.73 ± 0.27 | 0.04 ± 0.02 c | 23.3 ± 1.6 b | 0.21 ± 0.03 | 0.003 ± 0.001 c |
| Digestate | | 315.7 ± 30.3 a | 2.34 ± 0.28 | 0.19 ± 0.05 a | 33.3 ± 3.8 a | 0.29 ± 0.03 | 0.020 ± 0.006 a |
| Urea | | 224.9 ± 35.6 b | 2.22 ± 0.72 | 0.12 ± 0.03 b | 23.5 ± 4.0 b | 0.26 ± 0.08 | 0.011 ± 0.004 b |
| | 0 | 223.1 ± 16.4 | 1.73 ± 0.27 | 0.04 ± 0.02 b | 23.3 ± 1.6 | 0.21 ± 0.03 | 0.003 ± 0.001 b |
| | 50 | 257.1 ± 46.8 | 2.31 ± 0.64 | 0.14 ± 0.03 a | 27.0 ± 4.9 | 0.28 ± 0.07 | 0.013 ± 0.004 a |
| | 100 | 283.5 ± 67.3 | 2.25 ± 0.45 | 0.17 ± 0.07 a | 29.8 ± 7.8 | 0.27 ± 0.06 | 0.017 ± 0.008 a |

Means that do not share a letter are statistically significantly different from each other ($p < 0.05$).

*2.4. Statistical Analysis*

The GHG emissions ($CO_2$, $CH_4$, and $N_2O$), yields, and biomass were examined by analysis of variance (ANOVA). The Tukey HSD test with a critical *p* value lower than 0.05 was used as pairwise comparison to test the significance between different N-fertilization rates (0, 50, 100 kg N ha$^{-1}$) and types of fertilizer (digestate and urea). The statistical package used to run the ANOVA analysis was Minitab 19.

## 3. Results

*3.1. Meteorological Information*

The average temperature recorded during the growing season was 23.0 °C (Figure 2). July and August were the warmest months, with an average temperature of 26.1 °C. Average maximum and minimum temperatures were of 29.3 °C and 17.0 °C, with maximum and minimum absolute values of 39.5 °C and 4.2 °C, respectively. The total amount of precipitation was 197.2 mm, distributed over 28 precipitation events. Most of the precipitation was recorded in May (135.2 mm), with erratic rainfall from June to September.

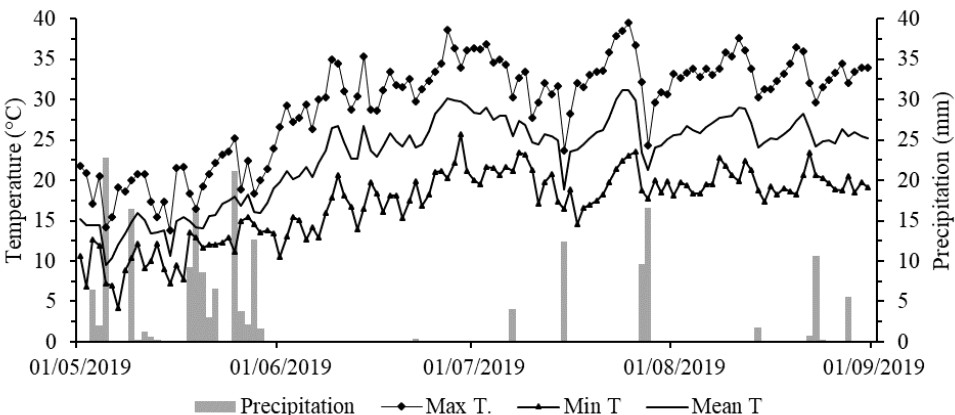

**Figure 2.** Average, maximum and minimum temperatures (°C), and precipitation (mm) observed during the growing season (May–August 2019).

*3.2. Carbon Dioxide Emissions*

Despite a high variability in the results, soil $CO_2$ emissions during the growing season showed similar trends in all treatments (Figure 3). Emissions increased following crop growth, peaking twice at each fertilization event (at 28 and 49 DAS, corresponding to stem elongation and flowering). Regardless of the N rate (50 and 100 kg N ha$^{-1}$), treatments with digestate produced higher emissions than synthetic fertilizers and the control (Table 3). Between the two fertilization events, and from M6 to the end of the experiment, soil $CO_2$ emissions experienced a notable decrease with time and, therefore, at harvesting (M9), soil $CO_2$ emissions were close to initial levels. Significantly higher ($p < 0.05$) cumulative $CO_2$ emissions were observed under 50D and 100D treatments (293.6 and 337.8 kg C ha$^{-1}$, respectively) than for the other treatments (Table 3). However, no significant emission differences ($p > 0.05$) were reported between the control, 50U and 100U (Table 3). Similar cumulative $CO_2$ emissions ($p < 0.05$) were recorded for the control and urea treatments, and these were approximately 30% lower to those of digestate (Table 3). Overall, changes in N fertilization rate (0, 50, and 100 kg N ha$^{-1}$) did not affect the $CO_2$ emission rate ($p > 0.05$) (Table 3).

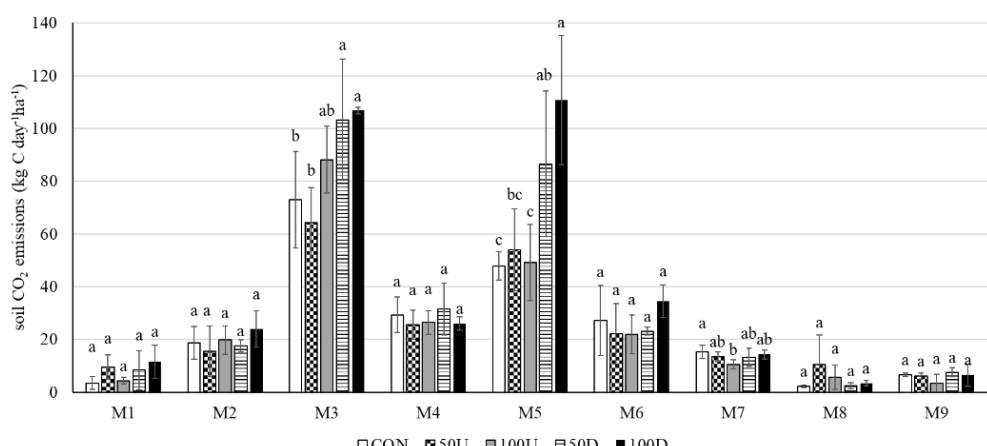

**Figure 3.** $CO_2$ emission fluxes (kg C $day^{-1}$ $ha^{-1}$) during the growing cycle for the control (CON), urea 50 kg N $ha^{-1}$ (50U), urea 100 kg N $ha^{-1}$ (100U), digestate 50 kg N $ha^{-1}$ (50D), and digestate 100 kg N $ha^{-1}$ (100D). Results that do not share letters are significantly different from each other ($p < 0.05$).

### 3.3. Methane Emissions

As for $CO_2$, two $CH_4$ emission peaks were observed in all treatments (M3 and M5), coinciding with each fertilization event (Figure 4). Nevertheless, during the second fertilization event (M5), emissions were considerably higher (0.9 to 1.2 kg C day $ha^{-1}$) to those observed during the first fertilization event (0.1 to 0.3 kg C day $ha^{-1}$). No significant $CH_4$ emission differences ($p > 0.05$) were displayed between treatments. However, during the second fertilization treatment, higher $CH_4$ emissions were observed in both digestate treatments, suggesting a marginal effect of water and methanogenic bacteria content on digestate. This was also confirmed by the cumulative $CH_4$ emissions, with higher values detected under 50D and 100D, though without displaying significant differences ($p > 0.05$) amongst treatments (Table 3). In M2 and M9, the soil acted as a $CH_4$ sink with low levels of $CH_4$ oxidation. Overall, during the growing season, no statistical differences ($p > 0.05$) were recorded for soil $CH_4$ emissions in all treatments, suggesting that $CH_4$ emissions were neither related to different N rates (0, 50, and 100 kg N $ha^{-1}$) nor to different types of fertilizer (digestate and urea) (Table 3).

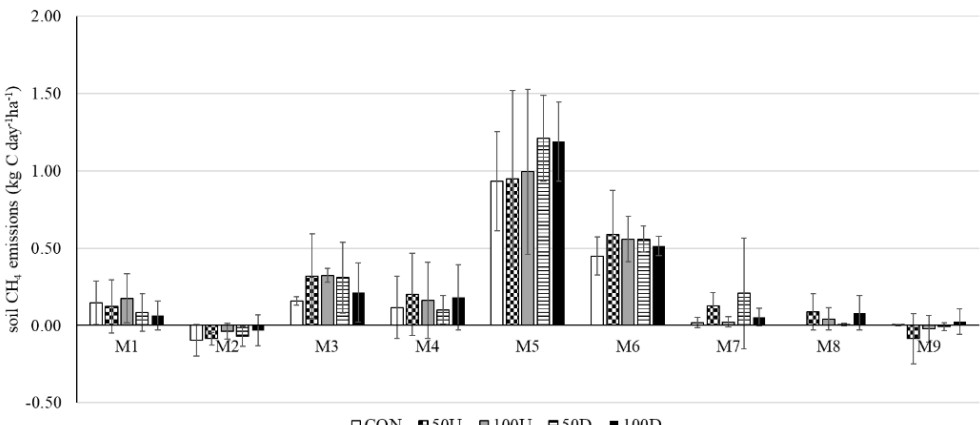

**Figure 4.** $CH_4$ emission fluxes (kg C $day^{-1}$ $ha^{-1}$) during the growing cycle for the control (CON), urea 50 kg N $ha^{-1}$ (50 U), urea 100 kg N $ha^{-1}$ (100 U), digestate 50 kg N $ha^{-1}$ (50 D), and digestate 100 kg N $ha^{-1}$ (100 D). Letters are not displayed since no statistical differences were observed between treatments ($p < 0.05$)).

### 3.4. Nitrous Oxide Emissions

For the $N_2O$ emissions fluxes from the soil, similar trends were detected to those of $CO_2$ and $CH_4$, with an $N_2O$ increase at each fertilization event (M3 and M5) and following measurement (M6) (Figure 5). Nevertheless, during the second fertilization event (M5), $N_2O$ emissions showed a different trend to that of $CO_2$ and $CH_4$, based on the type of fertilizer (digestate and urea) and N rate (50 and 100 kg N ha$^{-1}$). For the control, $N_2O$ emissions were not reported during the entire growing cycle, except for M3. Regardless of the amount of N (50 and 100 kg N ha$^{-1}$), urea had a longer emission lag-time than digestate and, therefore, higher $N_2O$ emissions were observed at M6 than at M5. In fact, digestate was more reactive than urea, showing an emission peak under 50D and 100D immediately after each fertilization event. Nevertheless, only significantly higher ($p < 0.05$) emissions were observed under 100D. As for $CO_2$ and $CH_4$, $N_2O$ emissions decreased between the two fertilization events, with emissions returning to initial levels at the end of the experiment. From the analysis of cumulative $N_2O$ emissions (Table 3), results showed higher $N_2O$ emissions ($p < 0.05$) at 100D compared to other treatments (CON and 50D). Significant differences ($p < 0.05$) were reported between 100D and 100U, 50U and 50D, as well as when comparing 100D, 100U, 50U, and 50D to the control. Regardless of the N supply, $N_2O$ emissions were significantly affected by the type of fertilizer ($p < 0.05$), with digestate producing the highest $N_2O$ emissions compared to urea. $N_2O$ emissions were affected by different N rates, being 100 kg N ha$^{-1}$ the treatment (100D and 100U) with highest $N_2O$ emissions.

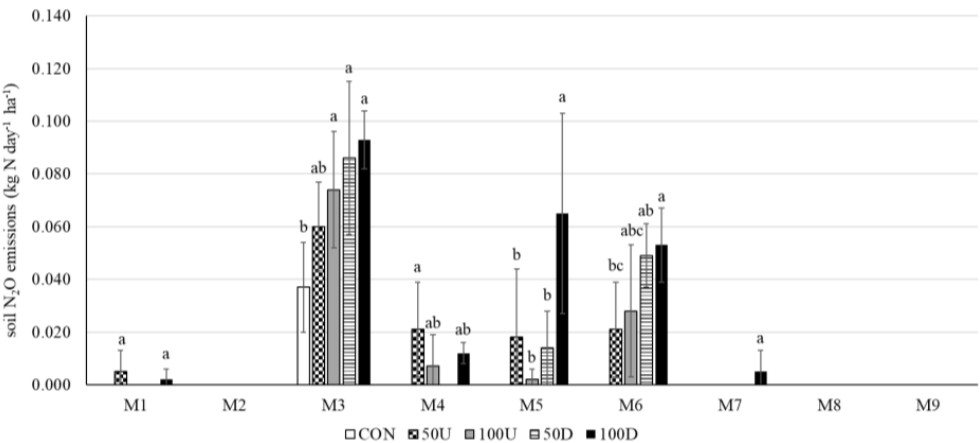

**Figure 5.** $N_2O$ emission fluxes (kg N day$^{-1}$ ha$^{-1}$) during the growing cycle for the control (CON), urea 50 kg N ha$^{-1}$ (50U), urea 100 kg N ha$^{-1}$ (100U), digestate 50 kg N ha$^{-1}$ (50D), and digestate 100 kg N ha$^{-1}$ (100D). Results that do not share letters are significantly different from each other ($p < 0.05$).

### 3.5. Quinoa Yields and Biomass Production

From the analysis of results, we observed no differences in yield and biomass neither between different N rates (0, 50, and 100 kg N ha$^{-1}$) nor types of fertilizers (urea and digestate) (Table 2). Even though the highest seed yields were obtained under 100U and 50U (894 and 852 kg ha$^{-1}$, respectively), no significant yield differences ($p > 0.05$) were found when comparing 100D, 50D, and the control. A similar behavior was observed with biomass production, with no significant differences between different N rates nor fertilizers. While the highest dried-biomass production was reported under 100U and 100D with 4064 and 3832 kg ha$^{-1}$, respectively, the lowest was found both under 50U and the control (3188 kg ha$^{-1}$). The average harvested index for all the treatments was 23.7%.

### 3.6. Yield-Scaled Emissions

The ratio between cumulative GHG emissions (Table 3) and observed yields (Table 2) were crucial for determining the environmental impacts of quinoa. Relative $CO_2$, $CH_4$, and $N_2O$ emissions, expressed as kg and g of C and N kg seed$^{-1}$ produced (depending on the gas), showed alike behavior in all treatments (CON, 50D, 100D, 50U, and 100U). For all three gases ($CO_2$, $CH_4$, and $N_2O$), values were highest at 100D followed by 50D, 100U, 50U, and, finally, the control. For $N_2O$, significant differences ($p < 0.05$) for relative emissions were observed under 100D when compared to the other treatments (Table 3). The gas-specific GWP for each GHG was used to calculate the yield-scaled emission (Table 4) for each treatment. Yield-scaled emissions were significantly higher for digestate (0.075 kg $CO_2$eq kg seed$^{-1}$) than for the control (0.013 kg $CO_2$ eq. kg seed$^{-1}$). Despite the different levels of N fertilization, no significant differences were reported between treatments. In addition, the gas-specific emissions per kg of seeds produced for different types of fertilizers showed that digestate had higher $N_2O$ emissions than urea.

**Table 4.** Yield-scaled emissions ($CO_2$, $CH_4$, and $N_2O$) per kg of quinoa seed produced and Global Warming Potential (GWP) (kg $CO_2$eq kg seed$^{-1}$) of each treatment.

| Fertilizer Type | N-Treatment (kg ha$^{-1}$) | $CO_2$ (kg C kg Seed$^{-1}$) | $CH_4$ (mg C kg Seed$^{-1}$) | $N_2O$ (mg N kg Seed$^{-1}$) | $CH_4$ (kg $CO_2$ eq kg Seed$^{-1}$) | $N_2O$ (kg $CO_2$ eq kg Seed$^{-1}$) | Total (kg $CO_2$ eq kg Seed$^{-1}$) |
|---|---|---|---|---|---|---|---|
| Control | 0 | 0.266 ± 0.024 | 2.1 ± 0.3 | 0.04 ± 0.02 b | 0.057 ± 0.007 | 0.013 ± 0.006 b | 0.337 ± 0.024 |
| Digestate | 50 | 0.402 ± 0.082 | 3.2 ± 0.2 | 0.21 ± 0.05 a | 0.090 ± 0.006 | 0.061 ± 0.016 a | 0.553 ± 0.100 |
| | 100 | 0.438 ± 0.097 | 3.0 ± 0.7 | 0.30 ± 0.06 a | 0.083 ± 0.021 | 0.089 ± 0.018 a | 0.609 ± 0.134 |
| Urea | 50 | 0.265 ± 0.076 | 2.7 ± 1.5 | 0.15 ± 0.05 ab | 0.076 ± 0.042 | 0.045 ± 0.016 ab | 0.386 ± 0.128 |
| | 100 | 0.301 ± 0.145 | 3.0 ± 1.9 | 0.15 ± 0.08 ab | 0.084 ± 0.052 | 0.044 ± 0.025 ab | 0.429 ± 0.213 |
| | Control | 0.266 ± 0.024 b | 2.1 ± 0.3 | 0.04 ± 0.02 b | 0.057 ± 0.007 | 0.013 ± 0.006 b | 0.337 ± 0.024 b |
| | Digestate | 0.420 ± 0.083 a | 3.1 ± 0.5 | 0.25 ± 0.07 a | 0.087 ± 0.014 | 0.075 ± 0.021 a | 0.581 ± 0.110 a |
| | Urea | 0.283 ± 0.105 ab | 2.9 ± 1.5 | 0.15 ± 0.06 b | 0.080 ± 0.043 | 0.045 ± 0.019 b | 0.408 ± 0.159 ab |
| | 0 | 0.266 ± 0.024 | 2.1 ± 0.3 | 0.04 ± 0.02 b | 0.057 ± 0.007 | 0.013 ± 0.006 b | 0.337 ± 0.024 |
| | 50 | 0.334 ± 0.103 | 3.0 ± 1.0 | 0.18 ± 0.06 ab | 0.083 ± 0.028 | 0.053 ± 0.017 ab | 0.470 ± 0.137 |
| | 100 | 0.370 ± 0.133 | 3.0 ± 1.3 | 0.22 ± 0.10 a | 0.084 ± 0.035 | 0.067 ± 0.031 a | 0.519 ± 0.187 |

Means that do not share a letter are statistically significantly different from each other ($p < 0.05$).

## 4. Discussion

### 4.1. Carbon Dioxide Emissions

Our results showed that $CO_2$ emissions peak at each fertilization event, afterwards, emissions declined in a similar way to that reported by [8,23]. The $CO_2$ emissions peaked in spite of the amount of N (50 and 100 kg N ha$^{-1}$) and type of fertilizer (urea and digestate), with similar findings as those observed in the control (0 kg N ha$^{-1}$). Hence, the increase in $CO_2$ emissions was not just attributed to fertilization treatments, but to agricultural practices, in particular, mechanical weeding. Since the control was subjected to the same agricultural practices as that of the other treatments, the peak in $CO_2$ displayed a similar behavior among treatments. Soil $CO_2$ emissions were produced during harrowing, releasing $CO_2$ trapped in the soil pores [24,25]. The destruction of soil aggregates, through harrowing, improved soil aeration and the incorporation of organic matter into the soil. As a result, microorganism activity was enhanced which led to a rapid decomposition of soil organic matter [25]. In addition, $CO_2$ emissions from agricultural soils were affected by a wide range of factors such as soil moisture, temperature, soil organic matter, and pH, among others, thereby rendering emission dynamics highly variable [8,26,27]. The compounded effect of all these factors resulted in a high variability in $CO_2$ emissions.

Furthermore, in this study, we observed higher $CO_2$ emissions under higher N rates of digestate (100D) than under lower N rates of both digestate (50D) and/or urea (100U and 50U). This was probably due to the combined effect of high-water and organic matter content in digestate, which uniformly reached the rhizosphere. Moreover, water played a key role in the development of soil microbial activity. For example, the action of water in

digestate, together with high atmospheric temperatures (Figure 1), created optimal microbial growing conditions, thereby increasing metabolic activity and enhancing microbial respiration, which consequently resulted in the release of $CO_2$. Similarly, organic matter in digestate also enhanced the soil microbial growth. Previous studies demonstrated that organic matter represented a fundamental source of carbon (C) for the metabolism of soil bacteria and, thus, further increased $CO_2$ emissions [28,29].

With regards to soil $CO_2$ emissions from urea, our results did not show significant differences on cumulative soil $CO_2$ emissions between the control, 50U, and 100U. Therefore, given that the soil $CO_2$ emissions from urea were triggered by hydrolysis (occurring under the presence of water and urease enzyme), and that the C contained in urea was lost by volatilization, no significant differences were reported among treatments.

### 4.2. Methane Emissions

The soil $CH_4$ emissions observed in this experiment were negligible compared to those of $CO_2$. Nevertheless, $CH_4$ emission trends were similar to those of $CO_2$, increasing after each fertilization event. Dried conditions during the growing cycle, intercalated by heavy rainfall events, were responsible for low $CH_4$ cumulative emissions (28.9 kg C ha$^{-1}$, average of all treatments). These values were consistent with the findings of [30,31], affirming that soil water and anaerobic conditions were the main drivers of $CH_4$ emissions. Despite the discrepancies found within the literature, study [32] concluded that N fertilization, and in particular $NH_4^+$, inhibit $CH_4$ oxidation due to its similar molecular size and the low specificity of the monooxygenase enzymes of methanotrophic bacteria. This competition was first described by study [33], demonstrating the inhibitory effect of $NH_4^+$ on $CH_4$ oxidation. As a result, the $CH_4$ emissions from digestate can be ascribed to the intrinsic content of methanogenic bacteria within digestate [34]. In addition, due to the combined effect of physical disturbances within the soil from harrowing, and the use of different types of fertilizers, $CH_4$ emissions from digestate were similar to those observed in urea and the control. The authors of [24,30] affirmed that land use changes, from natural conditions to agricultural lands, strongly reduced the $CH_4$ oxidation potential of soils, which switched from being a sink to become a $CH_4$ source. In this experiment, $CH_4$ absorption from the soil was observed at M2 and M9, when harrowing did not take place (Figure 4). Hence, tillage was the main factor responsible for $CH_4$ emissions. Tillage was previously shown to have a greater effect on $CH_4$ emissions than the type of fertilizers and N fertilization rates by directly disturbing the methane-oxidizing community at the soil level [35].

### 4.3. Nitrous Oxide Emissions

The present study revealed the effect of N rate on $N_2O$ emissions, with a positive relationship between increasing N rates and $N_2O$ emissions, mostly from digestate. Due to the shallowed root system of the selected quinoa genotype, less N was taken up by the root system and, therefore, higher residual N was left in the soil prone to environmental losses (soil leaching and volatilization). As for $CO_2$ and $CH_4$, $N_2O$ emissions followed a similar behavior during the growing season, with two emission peaks at each fertilization event. According to studies [29,36], $N_2O$ emissions were strongly related to soil moisture conditions and available N-compounds, mostly $NH_4^+$. During the first fertilization event, all treatments showed alike trend, though higher emissions under 100D due to a higher water (>98%), $NH_4^+$ (>75% of total N), and organic C content present in digestate. The compounded effect of all these factors accelerated soil $N_2O$ emissions. Relevant $N_2O$ emissions were also reported for 50D as well as for urea treatments. $N_2O$ emissions from the latter treatments were associated to heavy precipitation observed before applying fertilizers (Figure 2), which improved soil water content and enhanced urea decomposition. Urea had a high $NH_4^+$ content, but as a granular fertilizer, required more water to degrade [30]. Harrowing performed at M5 (43 DAS) disaggregated the soil and ensured homogenous infiltration of water from precipitation. For example, the 4 mm of rain observed at 68 DAS resulted in urea degradation. Thus, the late production of $N_2O$ emissions were, to some

extent, explained by the delayed response of urea during the second fertilization event. Similarly, other studies showed a strong correlation between soil water content and $N_2O$ emissions following the urea fertilization event, even if precipitation occurred several days after fertilization [37,38]. Moreover, a decrease in $N_2O$ emissions was observed after the last fertilization event, from M7 to the end of the experiment, and reaching zero at M8 (Figure 5). In agreement with the latter observations were the findings made by [4,36]. $N_2O$ emissions from the control were only observed at M3, probably due to the simultaneous effect of weather conditions, harrowing, and the intrinsic soil N content. In this line, study [29] showed a slight increase in $N_2O$ emissions when soil pores were filled with more than 60% of water. Study [4] indicated that the physical conditions of fertilizers were responsible for increasing $N_2O$ emissions by 23% from digestate when compared to urea. Moreover, the dried climatic conditions occurring after M6, resulted in $N_2O$ emission differences between digestate and urea. Hence, the type of fertilizer had an impact on the amount of $N_2O$ emissions, which was related to the soil water content, N compounds, and C concentration present in fertilizers. This was consistent with studies [39,40], who reported an increase of $NH_4^+$-compounds in the soil with increasing $N_2O$ emissions. Overall, the higher water content found on digestate played a key role on $N_2O$ emission dynamics and, therefore, showed a significantly higher $N_2O$ emissions than in urea.

From the analysis on cumulative $N_2O$ emissions, 100D produced the highest $N_2O$ emissions because of the higher $NH_4^+$, organic C, and water content found in digestate (Table 1). The key role of water was also observed in the cumulative $N_2O$ emissions from 100U, showing very little differences when compared to 50U and 50D (Table 3). Due to its intrinsic characteristics of high-water content, N-compounds, and C content, digestate generated higher $N_2O$ emissions than urea.

### 4.4. Quinoa Yields and Environmental Impacts

The herein study did not show statistically significant differences in terms of biomass and yield production, neither for different types of fertilizers (digestate and urea) and N fertilization levels (0, 50, and 100 kg N ha$^{-1}$). The soil N content was adequate to satisfy the N requirements of quinoa and, consequently, the additional N from fertilizers did not affect the final seed yield. Similar observations were made by studies [15–17,41], concluding that N requirements of quinoa were relatively low. However, the present results were discrepant to those of studies [10,19], which reported significant yield differences when applying higher amounts of poultry 1.7 Mg seed ha$^{-1}$ with 54 Mg ha$^{-1}$ poultry fertilizer) and slurry fertilizer (2.2 Mg ha$^{-1}$ with 180 kg N ha$^{-1}$). The findings of studies [12–14] were also in accordance with the latter research, showing a steady yield increase of up to 75–100 kg N ha$^{-1}$ and 120 kg N ha$^{-1}$. In addition, study [42] suggested that yields in south Italy were considerably lower (1.4 Mg ha$^{-1}$) if sowing in May rather than in April. In our experiment, the sowing was delayed to May due to adverse weather conditions of April 2020 and, therefore, lower yields (800 kg ha$^{-1}$) to those of study [42] were reported. This was due to high temperatures during the flowering stage (end of June) (Figure 2), which had a detrimental effect on seed pollination [43].

From the assessment of quinoa's environmental impacts, this study reported lower values (0.46 kg $CO_2$eq kg$^{-1}$ of quinoa seed, average of all treatments) to those of studies [19,44] in Peru (1.03 and 0.88 kg $CO_2$eq kg$^{-1}$ of quinoa harvested, respectively). While the latter study used a Life Cycle Assessment (LCA) software to estimate GHG emissions, our study was based on field observations. Lastly, the daily $CO_2$ emissions observed in our work (27.4 kg $CO_2$-C ha$^{-1}$ day$^{-1}$ kg seed$^{-1}$, average of all treatments) halved those of maize (40–60 kg $CO_2$-C ha$^{-1}$ day$^{-1}$), whereas those of $N_2O$ (0.132 kg $N_2O$-N ha$^{-1}$ average values of all treatments) were notably lower to those of maize (2–7 kg $N_2O$-N ha$^{-1}$ during the growing season) [4,25,27]. This was invariably the result of higher agricultural inputs (fertilizers, pesticides, water, etc.) of maize compared to those of quinoa.

## 5. Conclusions

The expansion of crops with a low environmental impact and high nutritional properties, coupled with low-impact agricultural strategies, are increasingly drawing scientific and public attention. In this study, we evaluated GHG emissions of quinoa cultivation using different types of fertilizer and N rates in central Italy, Tuscany. We observed that the direct soil GHG emissions of digestate are higher (30% for $CO_2$ and 40% for $N_2O$) than those of urea. Although we only evaluated the direct emissions from the soil, further impact assessments should consider the indirect emissions from the production and subsequent spreading of fertilizers. However, as a by-product of a renewable resource, digestate is assumed to be a zero-impact fertilizer during the production phase, besides being accepted as an effective strategy for reusing resources within a farm. The former is key when comparing to conventional fertilizers such as urea, which requires less harrowing but has higher production emissions. For this reason, a more in-depth analysis is recommended both in the direct and indirect emissions deriving from different types of fertilizers. To conclude, based on our observations, the low N requirements of quinoa masked the effect of fertilization. Therefore, the actual convenience of fertilization must be regularly evaluated according to the agroclimatic conditions of each site. This study represents a starting point for the definition of low-impact quinoa production with the prospects of developing more sustainable farming systems around the world.

**Author Contributions:** Conceptualization, L.V. and J.A.-B.; Methodology, L.V., J.A.-B. and R.V.; Formal Analysis, J.A.-B.; Writing, L.V. and J.A.-B.; Writing—Review and Editing, A.D.M. and S.O.; Supervision, A.D.M. and S.O. All authors have read and agreed to the published version of the manuscript.

**Funding:** This work was partially funded by the project SYSTEMIC "An integrated approach to the challenge of sustainable food systems adaptive and mitigatory strategies to address climate change and malnutrition", Knowledge hub on Nutrition and Food Security, in a joint action of JPI HDHL, JPI OCEANS and FACCE JPI launched in 2019 under the ERA NET ERA HDHL (n 696295).

**Institutional Review Board Statement:** Not applicable.

**Informed Consent Statement:** Not applicable.

**Data Availability Statement:** Not applicable.

**Acknowledgments:** The authors acknowledge Euro Pannacci from the Department of Agricultural, Food and Environmental Sciences (University of Perugia) for the provision of quinoa seed. Special recognitions to Azienda Agricola Marchese De' Frescobaldi—Fattoria di Corte for providing this research with digestate.

**Conflicts of Interest:** The authors declare no conflict of interest.

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
