# Peer review of "Greenhouse Gas Emissions and Yield Production from an Organic and Conventional Fertilization on Quinoa"

_agronomy, doi:10.3390/agronomy12051015_

Round 1
Reviewer 1 Report
The manuscript “The dilemma between yield enhancement and environmental impacts: assessment of greenhouse gas emissions on a quinoa field” quantifies and compares the effects of fertilizer type and amount on greenhouse gas emissions per area and per kg grain yield for quinoa cropping. The study is interesting, well written and fits into the scope of the journal. However there are some open questions and problems in the experimental approach that should be addressed/ discussed before publishing:
- Weekly measurements of N2O fluxes in the growing period might not describe N2O peaks after fertilisation events well enough
- N2O fluxes might occur due to crop residues after harvest. Therefore measuring just during crop growth would underestimate the N2O fluxes that should be attributed to Quinoa (same for CO2)
- It is not described what kind of CO2 flux is actually measured (crops in or outside the chambers)
- It is not clear that net Co2 fluxes could be quantified by those measurements described here, because respiration and gross primary productivity show a daily fluctuation while here CO2 is measured for 1 hr per day. Attribution of organic fertilizers or crop residues to the net CO2 gas exchange are not considered. Organic fertilisation with digestate should promote increasing SOC stocks with SOC stock changes showing if the agricultural system is acting as a CO2 sink or source.
- Because of the uncertainties of Co2 emission estimates it is not clear
- if the treatments can be compared and
- if CO2 equivalents of CO2, NO2 and CH4 emission measurements can be compared
L31 quite a high discrepancy between those 2 studies; maybe it could be described as some range, not clear for what region this is discussed, globally? Studies about national emissions even show emission factors below 1 ; also losses as N2O and N loss mixed in this sentence.
L86 Experiment duration May till August does not represent annual conditions with regard to N2O emissions. Maybe residual decomposition in autumn could cause N2O emissions as well. This would not be considered with this experimental approach.
L99 Maybe, information about pH and SOC could be added here
L125 /128 weekly measurements of N2O might not capture emission peaks after fertilisation at DAS 28 and 49
L125 “were recorded twice a week” Measuring 7,16, 28 days after seeding (DAS) is rather weekly
L132/138 were crops inside the chambers ?
Figure 3 caption: There are no letters in the figure showing significant differences
L272.. It is not clear from the MM section which kind of CO2 flow is actually measured (net balance of respiration and GPP, or just respiration). Beside this, respiration and photosynthesis follow daily cycles, with just respiration during the night. I doubt that weekly measurements with one hour closing time are adequate to estimate net CO2 fluxes.
L368 I suggest “was taken up” instead of “was uptake”
L406 It is not clear that water has a key role here. Higher emissions might just be caused by higher C and N inputs in 100D with increased mineralisation and O2 consumption
Line 443 I would suggest “In this study”
Author Response
Responses to Reviewer 1
The manuscript “The dilemma between yield enhancement and environmental impacts: assessment of greenhouse gas emissions on a quinoa field” quantifies and compares the effects of fertilizer type and amount on greenhouse gas emissions per area and per kg grain yield for quinoa cropping. The study is interesting, well written and fits into the scope of the journal. However, there are some open questions and problems in the experimental approach that should be addressed/ discussed before publishing:
- Weekly measurements of N2O fluxes in the growing period might not describe N2O peaks after fertilisation events well enough
We agree with the comment and generally measurements on N2O emissions should be more frequent. However, in dry conditions, as the present experiment, soil GHG fluxes are strongly reduced and biweekly measurements ensures to investigate the soil emissions fluxes [6]. Authors affirmed that, despite the positive effect of high temperatures on soil emission fluxes, low soil water content conditions strongly reduce the nutrients availability for microorganisms overcoming the temperature effect and thus, hampering emissions.
- N2O fluxes might occur due to crop residues after harvest. Therefore, measuring just during crop growth would underestimate the N2O fluxes that should be attributed to Quinoa (same for CO2)
We agree with reviewer on the role of crop residues on increasing GHG fluxes. However, in this experiment the entire plants were removed for yields (grain) and biomass (stems and leaves) production assessment (line 119-121, in Chapter 2.2). For this reason, we assumed that GHG fluxes after harvest were negligible compared to those produced during the production cycle as reported by Lehtinen et al. (2014).
- It is not described what kind of CO2 flux is actually measured (crops in or outside the chambers)
Thanks for the comment. In this experiment soil CO2 emissions were measured and crops were outside chambers. We included this information into the manuscript at line 147-148.
- It is not clear that net Co2 fluxes could be quantified by those measurements described here, because respiration and gross primary productivity show a daily fluctuation while here CO2 is measured for 1 hr per day. Attribution of organic fertilizers or crop residues to the net CO2 gas exchange are not considered. Organic fertilisation with digestate should promote increasing SOC stocks with SOC stock changes showing if the agricultural system is acting as a CO2 sink or source.
Measurement protocol was in accordance to [6] and [34] reporting the best time closure of chambers for GHG emissions monitoring. Moreover, performing measurements at mid-morning ensures obtaining representative data since this is the moment of the day when temperatures are similar to the daily average.
In this experiment we used the liquid fraction of digestate that was characterized by a low content of dry matter, and therefore of organic matter. For this reasons we have considered the contribution of digestate on SOC accumulation negligible.
- Because of the uncertainties of CO2 emission estimates it is not clear if the treatments can be compared and if CO2 equivalents of CO2, NO2 and CH4 emission measurements can be compared
We agree with reviewer and this is one limitation of the study. We described in both Results and Discussion chapters the CO2 emissions and hypothesized some explanation. As reported in Discussion, CO2 emissions are often influenced by chemical to biological and physical factors that, in some cases, can mask the effect of fertilization [26]. Nevertheless, we included additional mentions to highlight this issue (342-343).
L31 quite a high discrepancy between those 2 studies; maybe it could be described as some range, not clear for what region this is discussed, globally? Studies about national emissions even show emission factors below 1; also losses as N2O and N loss mixed in this sentence.
We have made corrections in order to make the text clearer
L86 Experiment duration May till August does not represent annual conditions with regard to N2O emissions. Maybe residual decomposition in autumn could cause N2O emissions as well. This would not be considered with this experimental approach.
We agree with reviewer that this approach does not allow investigating annual N2O fluxes. Nevertheless, the intent of this experiment was the assessment of GHG emission fluxes during the growing season of Quinoa. Moreover, as abovementioned, the entire plant removal for biomass production analysis greatly reduced the crop residues amount into the soil and the risk of post-harvest N2O peaks.
L99 Maybe, information about pH and SOC could be added here
We agree with reviewer that these information would be useful. However, in this experiment we have no data about these factors.
L125 /128 weekly measurements of N2O might not capture emission peaks after fertilisation at DAS 28 and 49
Reply to this comment is properly reported above
L125 “were recorded twice a week” Measuring 7,16, 28 days after seeding (DAS) is rather weekly
We apologize for the mistake, corrected into the text.
L132/138 were crops inside the chambers?
As reported in the reply above, plants were not included into the chambers in order to monitor just the soil GHG emission fluxes
Figure 3 caption: There are no letters in the figure showing significant differences
We apologize for the missing letters, corrected into the text.
L272 It is not clear from the MM section which kind of CO2 flow is actually measured (net balance of respiration and GPP, or just respiration). Beside this, respiration and photosynthesis follow daily cycles, with just respiration during the night. I doubt that weekly measurements with one-hour closing time are adequate to estimate net CO2 fluxes.
We explained above that in this experiment soil GHG fluxes were monitored (including CO2). Plants were not included in chambers since just soil emission fluxes were monitored. We included some specifications into the text.
L368 I suggest “was taken up” instead of “was uptake”
Corrected
L406 It is not clear that water has a key role here. Higher emissions might just be caused by higher C and N inputs in 100D with increased mineralisation and O2 consumption
We agree with reviewer’s comment. We deleted this sentence which was confusing
Line 443 I would suggest “In this study”
Corrected

Reviewer 2 Report
The manuscript ID 1658173 entitled” The dilemma between yield enhancement and environ mental impacts: assessment of greenhouse gas emissions on a quinoa field “. The authors failed to tackle the issues adequately. There are many issues were observed throughout the manuscript and shown below and at the attached reviewed manuscript with more detailed comments.
- Title should be changed to be clearer and cover the whole study and related to their treatments.
13 - It is not clear? Rewrite this sentence.
14- At the abstract, clarify what does mean each abbreviation.
25 Change the keywords with general words that can be related to your study but not like what it is here.
27 – introduction should be rewritten because it is not organized properly.
29 - What means IPCC? You should write what does it mean.
29 - This is a different statement.
32 - We knew that the N2O or No2 … etc, these gasses can be lost when no enough aeration into the soil that would increase the N reduction and getting the N gasses. This is not a complete statement.
45 - Here, you clarify this, but you should do it at the beginning of this section. Here you just say GHG.
47 - it is ambiguous sentence. rewrite this sentence. which kind of discrepancies do you mean?
50 - What do you mean? it is not clear. I don’t understand why did you write this here?
For now, what are you trying to say is not clear and why you write this. All should be rewritten.
63 - Researchers, farmers, and growers are splitting applications to reduce the N losses.
81 - This is not an objective. you have to rewrite it.
84 - This study should have the physicochemical properties table for the soil samplings at the pre-plant. Specially, N, C, S, organic matter and soil texture with sand, clay, sand percentage.
88 - RCBD, you have to write complete.
90 – you have to write the rates like this , 0 , 50 , and 100 ….etc everywhere in this study.
91 - how did you get 16 plots, is this for 3 reps or each rep? it is not clear . rewrite this procedure like other articles.
91 - what do you mean sizing +-4 , you should complete the plot size like how many for length and width.
91 - how many rows?
96 - What is the tractors name or type? how did you apply fertilizers using this machine? write it.
97 – I prefer that you replace the DAS numbers with the growing stages.
98 - write the machine name. When did you apply the fertilizers (date)?
99 - Put them on a table for the preplant soil analysis.
103 - You have to write the harvested methodology and the harvester’s name. Also, explain that how did you calculate the yield, biomass, and harvest index.
107 - How many plants had been harvested? you should write the exact number.
108 - DW for what? Should be cited.
108 - Is it a total yield? if so, you have to write it here. Which biomass you have got?
110 - Delete form g plant and keep kg ha-1
117 - urea does not have an organic carbon
117 - write this as an experimental design as a context, not here.
128 – biweekly
142 - As I mentioned before, you should write it like 0, 50 , 100 …
157 - Move it to the M&M section.
168 - Move it to the M&M section.
Better than using DAS you can use growing stages as vegetative growth names. like at 20s DAS , you can call it like leaf development or formation of side shoots stage. Do the same for all dates.
174 - Better than using DAS you can use growing stages as vegetative growth names. like at 20s DAS , you can call it like leaf development or formation of side shhots stage. Do the same for all dates.
- - The same above note.
189 - Figure 3. x-axis should be written. At the fig caption, remove all the abbreviation, and keep the whole names. You can write them below the fig.
192 - Where are these letters?
196 - Add the whole name at least once at the paragraph biggning.
214 - Do the same notes at fig 3.
231, 232 , 237, and 238 - write the whole names, e.g., D means digestate, so, write digestate and do the same with the others in the context.
245 – Do the same as figs 3 and 4.
252 - Rewrite this sentence.
263 - write the whole name.
264 – table 2 – Rewrite the N – treatments 0 , 50 , 100 kg / ha for both types everywhere.
271 - ?
284 – spacing.
285 - 0,50,and 100 ….etc
299 - How did you determine yield - scaled emissions?
299 – what does (GWP ) mean?
299 - Change numbers in scientific notation, e.g., 6 *10-5 and do the same for all numbers. Or, change the unites, e.g., using mg better than g.
321 - If the om is decomposed enough, there is no much moist in.
334 - How is C volatilized? the N gases should be lost.
337 – Delete it.
347 - Why it is anaerobic conditions? How did you know ?
349 - Urea fertilizer when will be converted NH3 to NO3 , if we assume that moist and bacteria will be good. So, your statement here is not true.
353 - Delete any word (may ) at the whole text and replace it with can or any expression that refers to your facts.
354 - How did you know this? Have you determined the physical properties?
366 - How did you know this fact? you haven’t analyzed N sources in the soils.
383 - What do you mean by urea degradation?
393 - What does harrowing mean ?
400 - If the soil condition is having a heavy irrigated or water conditions will increase the anaerobic bacteria which will convert NO3 to NO2 that will be lost as N gases.
405 - Regularly, the NH4 at the end of the season should be lost, absorbed by the plant, and the little amount still into the soil. However, it is a weak statement.
419 - Ton should be changed to Mg / ha everywhere in this study.
430 - Write the whole name and then write the abbreviations.
440 - Conclusion should be rewritten.
433 - To reduce or mitigate the emissions is not succeeded using such methods explained in this article.
444 - Each crop has a different absorbing method and amounts with different soil textures and other soil physiochemical properties that could help to understand what the emission is will be. in this study using just N rates from two unusual fertilizers, i.e., these fertilizers are not commonly used at all regions in the world.
445 - What you have got in this study is not enough to tell this recommendation.
454 - Rewrite these sentences. What you did write here is general info.
Please see the attachment other comments in the attachment and revise your manuscript.

Author Response
Response to reviewer 2
The manuscript ID 1658173 entitled” The dilemma between yield enhancement and environ mental impacts: assessment of greenhouse gas emissions on a quinoa field “. The authors failed to tackle the issues adequately. There are many issues were observed throughout the manuscript and shown below and at the attached reviewed manuscript with more detailed comments.
Title should be changed to be clearer and cover the whole study and related to their treatments.
Thanks for the suggestion. We proposed an alternative title
13 - It is not clear? Rewrite this sentence.
Thanks for the comment, we rewrote the sentence.
14- At the abstract, clarify what does mean each abbreviation.
Corrected
25 - Change the keywords with general words that can be related to your study but not like what it is here.
We would like to maintain the proposed keywords since we believe they provide an adequate overview of the subject matter of the study and in common within the subject of the discipline. However, we added 3 more keywords in order to improve the impact of the manuscript
27 – introduction should be rewritten because it is not organized properly.
Thanks for the suggestion. We improved the Introduction following the reviewer’s comments.
29 - What means IPCC? You should write what does it mean.
Corrected
Reviewer’s properly pointed out that there are several sources of N losses and suggested to describe them. We improved the sentence but we’d like to avoid including other N losses description (NH3 volatilization, N leaching, erosion etc.) since they are out of the topic of the manuscript. This could confuse the reader who would not find references to these factors in the article.
29 - This is a different statement.
Thanks for this point. We corrected the sentence.
32 - We knew that the N2O or No2 … etc, these gasses can be lost when no enough aeration into the soil that would increase the N reduction and getting the N gasses. This is not a complete statement.
We agree with reviewer and apologize. There was a typing error between “soil” and “solid”. We were referring to the different N2O emission risks between the use of liquid and SOLID organic fertilizers. We corrected the sentence.
45 - Here, you clarify this, but you should do it at the beginning of this section. Here you just say GHG.
Corrected
47 - it is ambiguous sentence. rewrite this sentence. which kind of discrepancies do you mean?
Discrepancies are linked to the effect of fertilization on yields and biomass production of quinoa. We corrected the sentence in order to make it clearer for readers.
50 - What do you mean? it is not clear. I don’t understand why did you write this here? For now, what are you trying to say is not clear and why you write this. All should be rewritten.
This part follows the previous. However, following reviewer’s comment we re-organized this part in order to make it clearer. We listed the available literature about the fertilization effects on quinoa. We initially reported articles who observed positive effects, then those observing negative or redundant effects. Finally, we concluded this part stating the open questions related to organic fertilization and yields and biomass production of Quinoa.
63 - Researchers, farmers, and growers are splitting applications to reduce the N losses.
We agree with reviewer that splitting N is a well-known strategy to reduce N losses in agriculture. However, here we are referring on Quinoa and the effect of N splitting to improve yields (amount and protein content). The cited article [9] observed that this strategy has no effect on these two parameters.
81 - This is not an objective. you have to rewrite it.
Corrected
84 - This study should have the physicochemical properties table for the soil samplings at the pre-plant. Specially, N, C, S, organic matter and soil texture with sand, clay, sand percentage.
Thanks for the suggestion. However, in the present study analysis on soil samplings before sowing were not carried out on C, S and organic matter. Data about soil texture and N content before sowing are reported at line 110-112
88 - RCBD, you have to write complete.
Corrected
90 – you have to write the rates like this , 0 , 50 , and 100 ….etc everywhere in this study.
Corrected
91 - how did you get 16 plots, is this for 3 reps or each rep? it is not clear . rewrite this procedure like other articles.
Corrected
91 - what do you mean sizing +-4 , you should complete the plot size like how many for length and width.
Corrected
91 - how many rows?
Corrected
96 - What is the tractors name or type? how did you apply fertilizers using this machine? write it.
Corrected
97 – I prefer that you replace the DAS numbers with the growing stages.
We have specified which phenological phases each DAS corresponds to
98 - write the machine name. When did you apply the fertilizers (date)?
We included the name of machine. Fertilization was carried out 28 and 49 days after sowing (line 107)
99 - Put them on a table for the preplant soil analysis.
Thanks for the suggestion, however since data are very few we prefer to present them in the text avoiding increasing number of tables
103 - You have to write the harvested methodology and the harvester’s name. Also, explain that how did you calculate the yield, biomass, and harvest index.
Corrected
107 - How many plants had been harvested? you should write the exact number.
Corrected
108 - DW for what? Should be cited.
Corrected
108 - Is it a total yield? if so, you have to write it here. Which biomass you have got?
Corrected
110 - Delete form g plant and keep kg ha-1
Corrected
117 - urea does not have an organic carbon
Corrected
117 - write this as an experimental design as a context, not here.
Corrected and properly reported, as suggested, at line 99-101
128 – biweekly
Corrected
142 - As I mentioned before, you should write it like 0, 50 , 100 …
Corrected
157 - Move it to the M&M section.
Thanks for the suggestion but in M&M we described the source of meteorological data (line 123-128). We believe that results from meteorological data collection should be presented here in the Results chapter
168 - Move it to the M&M section.
Same for the previous comment
174 - Better than using DAS you can use growing stages as vegetative growth names. like at 20s DAS, you can call it like leaf development or formation of side shhots stage. Do the same for all dates.
Thanks for the suggestion. We explained which phenological phases each DAS corresponds to in M&M and we included this specification into the text as suggested.
The same above note.
Corrected
189 - Figure 3. x-axis should be written. At the fig caption, remove all the abbreviation, and keep the whole names. You can write them below the fig.
Thanks for the suggestion but we’d like to maintain this version in order to have a concise legend
192 - Where are these letters?
We apologize for the missing letters. Corrected
196 - Add the whole name at least once at the paragraph biggning.
Thanks for the suggestion but according to journal guidelines the whole name should be defined the first time they appear in Abstract and Main text sections
214 - Do the same notes at fig 3.
Same for the previous comment
231, 232, 237, and 238 - write the whole names, e.g., D means digestate, so, write digestate and do the same with the others in the context.
We have specified which name each abbreviation corresponds to in the M&M following reviewer’s suggestion and accordingly to journal’s guidelines.
245 – Do the same as figs 3 and 4.
Same for the previous comment
252 - Rewrite this sentence.
Corrected
263 - write the whole name.
Corrected
264 – table 2 – Rewrite the N – treatments 0, 50, 100 kg / ha for both types everywhere.
Thanks for suggestion but units (kg N/ha) are reported in the first row of table. Perhaps repeating the units should be redundant
271 - ?
Cumulative emissions provide the total amount of each gas emitted during the reference period, in this manuscript the growing season of Quinoa. We added a comment in the M&M chapter to specify that.
284 – spacing.
Corrected
285 - 0,50,and 100 ….etc
Corrected
299 - How did you determine yield - scaled emissions?
Corrected
299 – what does (GWP) mean?
Corrected
299 - Change numbers in scientific notation, e.g., 6 *10-5 and do the same for all numbers. Or, change the unites, e.g., using mg better than g.
Corrected
321 - If the om is decomposed enough, there is no much moist in.
Due to the high water content of digestate (approximately 98% from Tab.1), soil moisture should increase compared to other treatments.
334 - How is C volatilized? the N gases should be lost.
Carbon is volatilized inside the hydrolysis process that transform urea – (NH2)2CO – into 2NH3 and CO2. The N volatilization losses are described in Chapter 4.3 as N2O since, in this experiment we didn’t have the opportunity to monitor NH3 emissions due to the intrinsic complexity for this measurements.
337 – Delete it.
Corrected
347 - Why it is anaerobic conditions? How did you know?
This was affirmed by cited literature (Le Mer and Roger, 2001) that observed these conditions at the soil level. This is our hypothesis since we didn’t measure the O2 concentration at the soil level.
349 - Urea fertilizer when will be converted NH3 to NO3, if we assume that moist and bacteria will be good. So, your statement here is not true.
We agree with reviewer that last step of urea degradation is NO3-. However, there is an intermediate step where N is under NH4+ form. Following we reported some literature supporting this statement:
Kumar, V. Wegenet R.J. (1984). Urease activity and kinetics of urea transformation in soils. Soil Science 137 (4), 263-269.
Watson, C. J., Miller, H., Poland, P., Kilpatrick, D. J., Allen, M. D. B., Garrett, I. M. K., Christianson, C. B. (1994). Soil properties and the ability of the urease inhibitor n-(n-butyl) thiophosphoric triamide (nbtpt) to reduce ammonia volatilization from surface-applied urea. Soil Biology and Biochemistry 26 (9), 1165-1171.
Witte, C. P. (2011). Urea metabolism in plants. Plant Science, 180(3), 431-438.
353 - Delete any word (may) at the whole text and replace it with can or any expression that refers to your facts.
Corrected
354 - How did you know this? Have you determined the physical properties?
We didn’t analyse the physical properties of soil. However, we did this hypothesis since relevant soil disturbance occurred after harrowing with the walking tractor. Moreover, since harrowing was carried out on all treatments, as mechanical weeding, we concluded that this would be an explanation of similar behaviours of different treatments.
366 - How did you know this fact? you haven’t analyzed N sources in the soils.
We reported that N2O emissions were correlated to increasing N amount because of the results we observed (Tab. 3). However, I agree with reviewer that the second sentence where we tried to explain the observations can be confusing. We corrected this part according to reviewer’s suggestions.
383 - What do you mean by urea degradation?
With this term we are referring to decomposition process of urea in plants-available forms. Following reviewer’s suggestion, we changed this word.
393 - What does harrowing mean?
Harrowing is a superficial mechanical practice (approximately 10-20 cm depth) that reduce soil aggregates sizes created e.g. with ploughing. It is widely adopted to refine sowing bed or, as in this case, as mechanical weeding.
400 - If the soil condition is having a heavy irrigated or water conditions will increase the anaerobic bacteria which will convert NO3 to NO2 that will be lost as N gases.
We agree with reviewer about the N transformation processes occurring in anaerobic conditions or high soil moisture levels. However, in this experiment the irrigation was not adopted (M&M) due to the high drought tolerance of Quinoa. In the manuscript we affirmed that soil water content has effect on N2O emissions but this water was provided by digestate (approximately 98% of water, Tab. 1) that wasn’t enough to create full anaerobic conditions, but according to our observations, probably enough to encourage N2O emissions.
405 - Regularly, the NH4 at the end of the season should be lost, absorbed by the plant, and the little amount still into the soil. However, it is a weak statement.
We agree with reviewer about the NH4+ fate at the end of the growing season. However, as abovementioned for another comment, the cumulative emissions were the sum of observed emissions during the growing season when NH4+ was almost fully available for plants, bacteria etc.
419 - Ton should be changed to Mg / ha everywhere in this study.
Corrected
430 - Write the whole name and then write the abbreviations.
Corrected
440 - Conclusion should be rewritten.
433 - To reduce or mitigate the emissions is not succeeded using such methods explained in this article.
This was a general statement. However, we deleted and improved the previous one.
444 - Each crop has a different absorbing method and amounts with different soil textures and other soil physiochemical properties that could help to understand what the emission is will be. in this study using just N rates from two unusual fertilizers, i.e., these fertilizers are not commonly used at all regions in the world.
We agree with reviewer that each crop has different absorbing method and amounts that can vary according to different factors (soil texture, soil organic matter content, CEC, pH, climatic conditions etc.). For this reason, we specified that our observations are true in the present conditions in Tuscany.
Fertilizers adopted in this study are actually wide spread in many parts of the world. Urea is probably the most used fertilizer, mainly due to the high N content, despite the strong price increase following the dramatic war situation in Ukraine. Following the strong development of biogas technology, financial supports to produce renewable energy at circular economy level and the increased attention of world population to the environmental topic, the use of digestate as alternative fertilizer is well confirmed, at least in developed countries. Below we report our article from 2019 where we reported the development of biogas, and digestate use as fertilizer, in Europe:
Verdi, L., Kuikman, P. J., Orlandini, S., Mancini, M., Napoli, M., & Dalla Marta, A. (2019). Does the use of digestate to replace mineral fertilizers have less emissions of N2O and NH3?. Agricultural and Forest Meteorology, 269, 112-118.
445 - What you have got in this study is not enough to tell this recommendation.
Thanks for the suggestion. We changed this sentence reporting that from our observations only the direct soil GHG emissions from digestate are higher to those of urea.
454 - Rewrite these sentences. What you did write here is general info.
Thanks for the suggestion. We changed this part focusing on our observed results.
